# Learning to Navigate in Cities Without a Map

**Piotr Mirowski, Matthew Koichi Grimes, Mateusz Malinowski, Karl Moritz Hermann,**
**Keith Anderson, Denis Teplyashin, Karen Simonyan, Koray Kavukcuoglu,**
**Andrew Zisserman, Raia Hadsell**
DeepMind
London, United Kingdom
`{piotrmirowski, mkg, mateuszm, kmh, keithanderson, }@google.com`
`{teplyashin, simonyan, korayk, zisserman, raia}@google.com`

## Abstract

Navigating through unstructured environments is a basic capability of intelligent creatures, and thus is of fundamental interest in the study and development of artificial intelligence. Long-range navigation is a complex cognitive task that relies on developing an internal representation of space, grounded by recognisable landmarks and robust visual processing, that can simultaneously support continuous self-localisation ("I am *here*") and a representation of the goal ("I am going *there*"). Building upon recent research that applies deep reinforcement learning to maze navigation problems, we present an end-to-end deep reinforcement learning approach that can be applied on a city scale. Recognising that successful navigation relies on integration of general policies with locale-specific knowledge, we propose a dual pathway architecture that allows locale-specific features to be encapsulated, while still enabling transfer to multiple cities. A key contribution of this paper is an interactive navigation environment that uses Google Street View for its photographic content and worldwide coverage. Our baselines demonstrate that deep reinforcement learning agents can learn to navigate in multiple cities and to traverse to target destinations that may be kilometres away. The project webpage `http://streetlearn.cc` contains a video summarizing our research and showing the trained agent in diverse city environments and on the transfer task, the form to request the StreetLearn dataset and links to further resources. The StreetLearn environment code is available at `https://github.com/deepmind/streetlearn`.

## 1   Introduction

The subject of navigation is attractive to various research disciplines and technology domains alike, being at once a subject of inquiry from the point of view of neuroscientists wishing to crack the code of grid and place cells [2, 12], as well as a fundamental aspect of robotics research. The majority of algorithms involve building an explicit map during an exploration phase and then planning and acting via that representation. In this work, we are interested in pushing the limits of end-to-end deep reinforcement learning for navigation by proposing new methods and demonstrating their performance in large-scale, real-world environments. Just as humans can learn to navigate a city without relying on maps, GPS localisation, or other aids, it is our aim to show that a neural network agent can learn to traverse entire cities using only visual observations. In order to realise this aim, we designed an interactive environment that uses the images and underlying connectivity information from Google Street View, and propose a dual pathway agent architecture that can navigate within the environment (see Fig. 1a).

Learning to navigate directly from visual inputs has been shown to be possible in some domains, by using deep reinforcement learning (RL) approaches that can learn from task rewards – for instance, navigating to a destination. Recent research has demonstrated that RL agents can learn to navigate

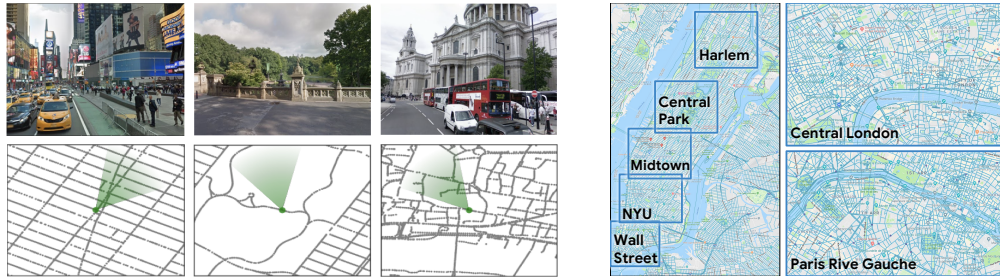

(a) Diverse views and corresponding local maps in Street View.  (b) Street View regions used in this study.

Figure 1: **(a)** Our environment is built of real-world places from Street View (we illustrate Times Square and Central Park in New York City and St. Paul's Cathedral in London). The green cone represents the agent's location and orientation. **(b)** We use large regions of London and Paris and in New York we focus on 5 different regions to show transfer.

house scenes [45, 42], mazes (e.g. [33]), and 3D games (e.g. [30]). These successes notwithstanding, deep RL approaches are notoriously data inefficient and sensitive to perturbations of the environment, and are more well-known for their successes in games and simulated environments than in real-world applications. It is therefore not obvious that they can be used for large-scale visual navigation based on real-world images, and hence this is the subject of our investigation.

The primary contributions of this paper are (a) to present a new RL challenge that features real world visual navigation through city-scale environments, and (b) to propose a modular, goal-conditional deep RL algorithm that can solve this task, thus providing a strong baseline for future research. *StreetLearn*[1] is a new interactive environment for reinforcement learning that features real-world images as agent observations, with real-world grounded content that is built on top of the publicly available Google Street View. Within this environment we have developed a traversal task that requires that the agent navigates from goal to goal within London, Paris and New York City.

To evaluate the feasibility of learning in such an environment, we propose an agent that learns a goal-dependent policy with a dual pathway, modular architecture with similarities to the interchangeable task-specific modules approach from [13], and the target-driven visual navigation approach of [45]. The approach features a recurrent neural architecture that supports both locale-specific learning as well as general, transferable navigation behaviour. Balancing these two capabilities is achieved by separating a recurrent neural pathway from the general navigation policy of the agent. This pathway addresses two needs. First, it receives and interprets the current goal given by the environment, and second, it encapsulates and memorises the features and structure of a single city region. Thus, rather than using a map, or an external memory, we propose an architecture with two recurrent pathways that can effectively address a challenging navigation task in a single city as well as transfer to new cities or regions by training only a new locale-specific pathway.

## 2 Related Work

Reward-driven navigation in a real-world environment is related to research in various areas of deep learning, reinforcement learning, navigation and planning.

**Learning from real-world imagery.** Localising from only an image may seem impossible, but humans can integrate visual cues to geolocate a given image with surprising accuracy, motivating machine learning approaches. For instance, convolutional neural networks (CNNs) achieve competitive scores on the geolocation task [41] and CNN+LSTM architectures improve on this [15, 31]. Several methods [5, 28], including DeepNav [6], use datasets collected using Street View or Open Street Maps and solve navigation-related tasks using supervision. RatSLAM demonstrates localisation and path planning over long distances using a biologically-inspired architecture [32]. The aforementioned methods rely on supervised training with ground truth labels: with the exception of the compass, we do not provide labels in our environment.

**Deep RL methods for navigation.** Many RL-based approaches for navigation rely on simulators which have the benefit of features like procedurally generated variations but tend to be visually simple and unrealistic [3, 26, 39]. To support sparse reward signals in these environments, recent navigational agents use auxiliary tasks in training [33, 25, 30]. Other methods learn to predict future measurements or to follow simple text instructions [16, 23, 22, 11]; in our case, the goal is designated using proximity to local landmarks. Deep RL has also been used for active localisation [10]. Similar to our proposed architecture, [45] show goal-conditional indoor navigation with a simulated robot and environment.

To bridge the gap between simulation and reality, researchers have developed more realistic, higher-fidelity simulated environments [17, 29, 38, 42]. However, in spite of their increasing photo-realism, the inherent problems of simulated environments lie in the limited diversity of the environments and the antiseptic quality of the observations. Photographic environments have been used to train agents on short navigation problem in indoor scenes with limited scale [9, 1, 7, 35]. Our real-world dataset is diverse and visually realistic, comprising scenes with vegetation, pedestrians or vehicles, diverse weather conditions and covering large geographic areas. However, we note that there are obvious limitations of our environment: it does not contain dynamic elements, the action space is necessarily discrete as it must jump between panoramas, and the street topology cannot be arbitrarily altered.

**Deep RL for path planning and mapping.** Several recent approaches have used memory or other explicit neural structures to support end-to-end learning of planning or mapping. These include Neural SLAM [44] that proposes an RL agent with an external memory to represent an occupancy map and a SLAM-inspired algorithm, Neural Map [36] which proposes a structured 2D memory for navigation, Memory Augmented Control Networks [27], which uses a hierarchical control strategy, and MERLIN, a general architecture that achieves superhuman results in novel navigation tasks [40]. Other work [8, 10] explicitly provides a global map that is input to the agent. The architecture in [21] uses an explicit neural mapper and planner for navigation tasks as well as registered pairs of landmark images and poses. Similar to [20, 44], they use extra memory that represents the ego-centric agent position. Another recent work proposes a graph network solution [37]. The focus of our paper is to demonstrate that simpler architectures can explore and memorise very large environments using target-driven visual navigation with a goal-conditional policy.

## 3   Environment

This section presents an interactive environment, named *StreetLearn*, constructed using Google Street View, which provides a public API[2]. Street View provides a set of geolocated $360°$ panoramic images which form the nodes of an undirected graph. We selected a number of large regions in New York City, Paris and London that contain between 7,000 and 65,500 nodes (and between 7,200 and 128,600 edges, respectively), have a mean node spacing of 10m, and cover a range of up to 5km (see Fig. 1b). We do not simplify the underlying connectivity, thus there are congested areas with complex occluded intersections, tunnels and footpaths, and other ephemera. Although the graph is used to construct the environment, the agent only sees the raw RGB images (see Fig. 1a).

### 3.1   Agent Interface and the Courier Task

An RL environment needs to specify the start space, observations, and action space of the agent as well as the task reward. The agent has two inputs: the image $\mathbf{x_t}$, which is a cropped, $60°$ square, RGB image that is scaled to $84 \times 84$ pixels (i.e. not the entire panorama), and the goal description $g_t$. The action space is composed of five discrete actions: "slow" rotate left or right ($\pm 22.5°$), "fast" rotate left or right ($\pm 67.5°$), or move forward—this action becomes a `noop` if there is not an edge in view from the current agent pose. If there are multiple edges in the view cone of the agent, then the most central one is chosen.

There are many options for how to specify the goal to the agent, from images to agent-relative directions, to text descriptions or addresses. We choose to represent the current goal in terms of its proximity to a set $\mathcal{L}$ of fixed landmarks: $\mathcal{L} = \{(Lat_k, Long_k)\}_k$, specified using the Lat/Long (latitude and longitude) coordinate system. To represent a goal at $(Lat_t^g, Long_t^g)$ we take a softmax over the distances to the $k$ landmarks (see Fig. 2a), thus for distances $\{d_{t,k}^g\}_k$ the goal vector

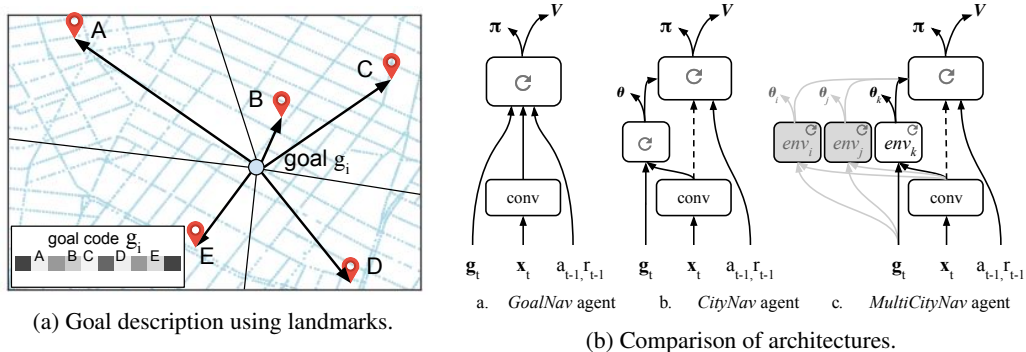

(a) Goal description using landmarks.

a. *GoalNav* agent    b. *CityNav* agent    c. *MultiCityNav* agent

(b) Comparison of architectures.

Figure 2: **(a)** In the illustration of the goal description, we show a set of 5 nearby landmarks and 4 distant ones; the code $g_i$ is a vector with a softmax-normalised distance to each landmark. **(b)** *Left:* GoalNav is a convolutional encoder plus policy LSTM with goal description input. *Middle:* CityNav is a single-city navigation architecture with a separate goal LSTM and optional auxiliary heading ($\theta$). *Right:* MultiCityNav is a multi-city architecture with individual goal LSTM pathways for each city.

contains $g_{t,i} = \exp(-\alpha d_{t,i}^g)/\sum_k \exp(-\alpha d_{t,k}^g)$ for the $ith$ landmark with $\alpha = 0.002$ (which we chose through cross-validation). This forms a goal description with certain desirable qualities: it is a scalable representation that extends easily to new regions, it does not rely on any arbitrary scaling of map coordinates, and it has intuitive meaning—humans and animals also navigate with respect to fixed landmarks. Note that landmarks are fixed per map and we used the same list of landmarks across all experiments; $g_t$ is computed using the distance to all landmarks, but by feeding these distances through a non-linearity, the contribution of distant landmarks is reduced to zero. In the Supplementary material, we show that the locally-continuous landmark-based representation of the goal performs as well as the linear scalar representation ($Lat_t^g, Long_t^g$). Since the landmark-based representation performs well while being independent of the coordinate system and thus more scalable, we use this representation as canonical. Note that the goal description is not relative to the agent's position and only changes when a new goal is sampled. Locations of the 644 manually defined landmarks in New York, London and Paris are given in the Supplementary material, where we also show that the density of landmarks does not impact the agent performance.

In the *courier* task, which we define as the problem of navigating to a series of random locations in a city, the agent starts each episode from a randomly sampled position and orientation. If the agent gets within 100m of the goal (approximately one city block), the next goal is randomly chosen and input to the agent. Each episode ends after 1000 agent steps. The reward that the agent gets upon reaching a goal is proportional to the shortest path between the goal and the agent's position when the goal is first assigned; much like a delivery service, the agent receives a higher reward for longer journeys. Note that we do not reward agents for taking detours, but rather that the reward in a given level is a function of the optimal distance from start to goal location. As the goals get more distant during the training curriculum, per-episode reward statistics should ideally reach and stay at a plateau performance level if the agent can equally reach closer and further goals.

## 4 Methods

We formalise the learning problem as a Markov Decision Process, with state space $\mathcal{S}$, action space $\mathcal{A}$, environment $\mathcal{E}$, and a set of possible goals $\mathcal{G}$. The reward function depends on the current goal and state: $R : \mathcal{S} \times \mathcal{G} \times \mathcal{A} \rightarrow \mathbb{R}$. The usual reinforcement learning objective is to find the policy that maximises the expected return defined as the sum of discounted rewards starting from state $s_0$ with discount $\gamma$. In this navigation task, the expected return from a state $s_t$ also depends on the series of sampled goals $\{g_k\}_k$. The policy is a distribution over actions given the current state $s_t$ and the goal $g_t$: $\pi(a|s,g) = Pr(a_t = a|s_t = s, g_t = g)$. We define the value function to be the expected return for the agent that is sampling actions from policy $\pi$ from state $s_t$ with goal $g_t$: $V^\pi(s,g) = E[R_t] = E[\sum_{k=0}^{\infty} \gamma^k r_{t+k}|s_t = s, g_t = g]$.

We hypothesise the courier task should benefit from two types of learning: general, and locale-specific. A navigating agent not only needs an internal representation that is general, to support cognitive

processes such as scene understanding, but also needs to organise and remember the features and structures that are unique to a place. Therefore, to support both types of learning, we focus on neural architectures with multiple pathways.

## 4.1 Architectures

The policy and the value function are both parameterised by a neural network which shares all layers except the final linear outputs. The agent operates on raw pixel images $x_t$, which are passed through a convolutional network as in [34]. A Long Short-Term Memory (LSTM) [24] receives the output of the convolutional encoder as well as the past reward $r_{t-1}$ and previous action $a_{t-1}$. The three different architectures are described below. Additional architectural details are given in the Supplementary Material.

The baseline **GoalNav** architecture (Fig. 2ba) has a convolutional encoder and *policy LSTM*. The key difference from the canonical A3C agent [34] is that the goal description $g_t$ is input to the policy LSTM (along with the previous action and reward).

The **CityNav** architecture (Fig. 2bb) combines the previous architecture with an additional LSTM, called the *goal LSTM*, which receives visual features as well as the goal description. The CityNav agent also adds an auxiliary heading ($\theta$) prediction task on the outputs of the goal LSTM.

The **MultiCityNav** architecture (Fig. 2bc) extends the CityNav agent to learn in different cities. The remit of the goal LSTM is to encode and encapsulate locale-specific features and topology such that multiple pathways may be added, one per city or region. Moreover, after training on a number of cities, we demonstrate that the convolutional encoder and the policy LSTM become general enough that only a new goal LSTM needs to be trained for new cities, a benefit of the modular approach [13].

Figure 2b illustrates that the goal descriptor $g_t$ is not seen by the policy LSTM but only by the locale-specific LSTM in the **CityNav** and **MultiCityNav** architectures (the baseline **GoalNav** agent has only one LSTM, so we directly input $g_t$). This separation forces the locale-specific LSTM to interpret the absolute goal position coordinates, with the hope that it then sends relative goal information (*directions*) to the policy LSTM. This hypothesis is tested in section 2.3 of the supplementary material.

As shown in [25, 33, 16, 30], auxiliary tasks can speed up learning by providing extra gradients as well as relevant information. We employ a very natural auxiliary task: the prediction of the agent's heading $\theta_t$, defined as an angle between the north direction and the agent's pose, using a multinomial classification loss on binned angles. The optional heading prediction is an intuitive way to provide additional gradients for training the convnet. The agent can learn to navigate without it, but we believe that heading prediction helps learning the geometry of the environment; the Supplementary material provides a detailed architecture ablation analysis and agent implementation details.

To train the agents, we use IMPALA [18], an actor-critic implementation that decouples acting and learning. In our experiments, IMPALA results in similar performance to A3C [34]. We use 256 actors for *CityNav* and 512 actors for *MultiCityNav*, with batch sizes of 256 or 512 respectively, and sequences are unrolled to length 50.

## 4.2 Curriculum Learning

Curriculum learning gradually increases the complexity of the learning task by presenting progressively more difficult examples to the learning algorithm [4, 19, 43]. We use a curriculum to help the agent learn to find increasingly distant destinations. Similar to RL problems such as Montezuma's Revenge, the courier task suffers from very sparse rewards; unlike that game, we are able to define a natural curriculum scheme. We start by sampling each new goal to be within 500m of the agent's position (phase 1). In phase 2, we progressively grow the maximum range of allowed destinations to cover the full graph (3.5km in the smaller New York areas, or 5km for central London or Paris).

## 5 Results

In this section, we demonstrate and analyse the performance of the proposed architectures on the courier task. We first show the performance of our agents in large city environments, next their

generalisation capabilities on a held-out set of goals. Finally, we investigate whether the proposed approach allows transfer of an agent trained on a set of regions to a new and previously unseen region.

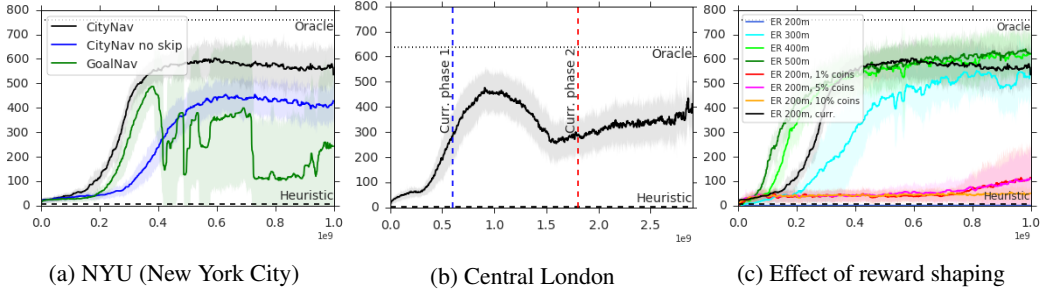

(a) NYU (New York City)  (b) Central London  (c) Effect of reward shaping

Figure 3: Average per-episode rewards (y axis) are plotted vs. learning steps (x axis) for the courier task. We compare the *GoalNav* agent, the *CityNav* agent, and the *CityNav* agent without skip connection on the NYU environment **(a)**, and the *CityNav* agent in London **(b)**. We also give *Oracle* performance and a *Heuristic* agent. A curriculum is used in London—we indicate the end of phase 1 (up to 500m) and the end of phase 2 (5000m). **(c)** Results of the *CityNav* agent on NYU, comparing radii of early rewards (ER) vs. ER with random coins vs. curriculum with ER 200m and no coins.

## 5.1 Courier Navigation in Large, Diverse City Environments

We first show that the *CityNav* agent, trained with curriculum learning, succeeds in learning the courier task in New York, London and Paris. We replicated experiments with 5 random seeds and plot the mean and standard deviation of the reward statistic throughout the experimental results. Throughout the paper, and for ease of comparison with experiments that include reward shaping, we report only the rewards at the goal destination (*goal rewards*). Figure 3 compares different agents and shows that the CityNav architecture with the dual LSTM pathways and the heading prediction task attains a higher performance and is more stable than the simpler GoalNav agent. We also trained a CityNav agent without the skip connection from the vision layers to the policy LSTM. While this hurts the performance in single-city training, we consider it because of the multi-city transfer scenario (see Section 5.4) where funeling all visual information through the locale-specific LSTM seems to regularise the interface between the goal LSTM and the policy LSTM. We also consider two baselines which give lower (*Heuristic*) and upper (*Oracle*) bounds on the performance. *Heuristic* is a random walk on the street graph, where the agent turns in a random direction if it cannot move forward; if at

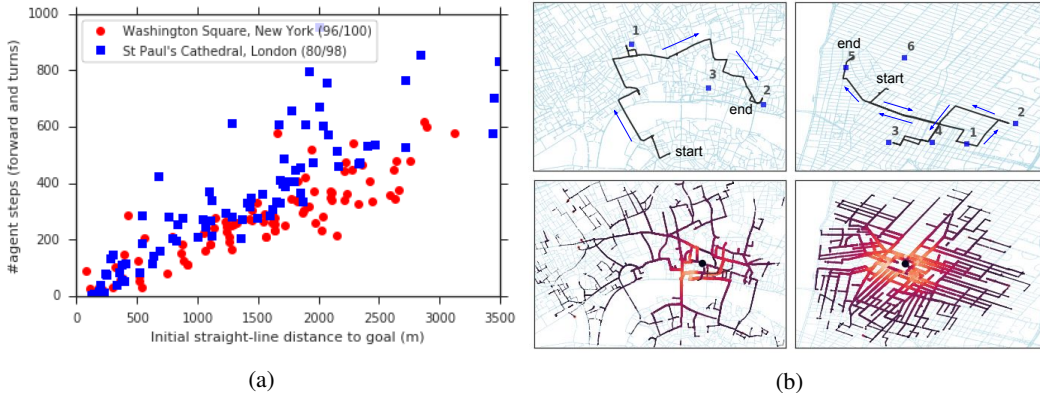

(a)  (b)

Figure 4: **(a)** Number of steps required for the *CityNav* agent to reach a goal from 100 start locations vs. the straight-line distance to the goal in metres. **(b)** *CityNav* performance in London (left panes) and NYU (right panes). *Top*: examples of the agent's trajectory during one 1000-step episode, showing successful consecutive goal acquisitions. The arrows show the direction of travel of the agent. *Bottom*: We visualise the agent's value function over 100 trajectories with random starting points and the same goal. Thicker and warmer colour lines correspond to higher value functions.

an intersection it will turn with a probability $p = 0.95$. *Oracle* uses the full graph to compute the optimal path using breath-first search.

We visualise trajectories from the trained agent over two 1000 step episodes (Fig. 4b (top row)). In London, we see that the agent crosses a bridge to get to the first goal, then travels to goal 2, and the episode ends before it can reach the third goal. Figure 4b (bottom row) shows the value function of the agent as it repeatedly navigates to a chosen destination (respectively, St Paul's Cathedral in London and Washington Square in New York).

To understand whether the agent has learned a policy over the full extent of the environment, we plot the number of steps required by the agent to get to the goal. As the number grows linearly with the straight-line distance to that goal, this result suggests that the agent has successfully learnt the navigation policy on both cities (Fig. 4a).

## 5.2   Impact of Reward Shaping and Curriculum Learning

To better understand the environment, we present further experiments on reward, curriculum. Additional analysis, including architecture ablations, the robustness of the agent to the choice of goal representations, and position and goal decoding, are presented in the Supplementary Material.

Our navigation task assigns a goal to the agent; once the agent successfully navigates to the goal, a new goal is given to the agent. The long distance separating the agent from the goal makes this a difficult RL problem with sparse rewards. To simplify this challenging task, we investigate giving early rewards (*reward shaping*) to the agent before it reaches the goal (we define goals with a 100m radius), or to add random rewards (*coins*) to encourage exploration [3, 33]. Figure 3c suggests that *coins* by themselves are ineffective as our task does not benefit from wide explorations. At the same time, large radii of reward shaping help as they greatly simplify the problem. We prefer curriculum learning to reward shaping on large areas because the former approach keeps agent training consistent with its experience at test time and also reduces the risk of learning degenerate strategies such as ascending the gradient of increasing rewards to reach the goal, rather than learn to read the goal specification $g_t$.

As a trade-off between task realism and feasibility, and guided by the results in Fig. 3c, we decide to keep a small amount of reward shaping (200m away from the goal) combined with curriculum learning. The specific reward function we use is: $r_t = \max(0, \min(1, (d_{ER} - d_t^g)/100)) \times r^g$, where $d_t^g$ is the distance from the current position of the agent to the goal, $d_{ER} = 200$ and $r^g$ is the reward that the agent will receive if it reaches the goal. Early rewards are given only once per panorama / node, and only if the distance $d_t^g$ to the goal is decreasing (in order to avoid the agent developing a behavior of harvesting early rewards around the goal rather than going directly towards the goal).

We choose a curriculum that starts by sampling the goal within a radius of 500m from the agent's location, and progressively grows that disc until it reaches the maximum distance an agent could travel within the environment (e.g., 3.5km, and 5km in the NYU and London environments respectively) by the end of the training. Note that this does not preclude the agent from going astray in the opposite direction several kilometres away from the goal, and that the goal may occasionally be sampled close to the agent. Hence, our curriculum scheme naturally combines easy with difficult cases [43], with the latter becoming more common over the period of time.

## 5.3   Generalization on Held-out Goals

Navigation agents should, ideally, be able to generalise to unseen environments [14]. While the nature of our courier task precludes zero-shot navigation in a new city without retraining, we test the *CityNav* agent's ability to exploit local linearities of the goal representation to handle unseen goal locations. We mask $25\%$ of the possible goals and train on the remaining ones (Fig. 5). At test time we evaluate the agent only on its ability to reach goals in the held-out areas. Note that the agent is still able to traverse *through* these areas, it just never samples a goal there. More precisely, the held-out areas are squares sized $0.01°$, $0.005°$ or $0.0025°$ of latitude and longitude (roughly 1km×1km, 0.5km×0.5km and 0.25km×0.25km). We call these grids respectively *coarse* (with few and large held-out areas), *medium* and *fine* (with many small held-out areas).

In the experiments, we train the *CityNav* agent for 1B steps, and next freeze the weights of the agent and evaluate its performance on held-out areas for 100M steps. Table 1 shows decreasing

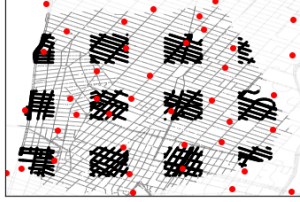

Figure 5: Illustration of medium-sized held-out grid with gray corresponding to training destinations, black corresponding to held-out test destinations. Landmark locations are marked in red.

| Grid Size | Train Rew | Test Rew | Test Fail | $T_{\frac{1}{2}}$ |
|---|---|---|---|---|
| FINE | 655 | 567 | 11% | 229 |
| MEDIUM | 637 | 293 | 20% | 184 |
| COARSE | 623 | 164 | 38% | 243 |

Table 1: *CityNav* agent generalization performance (reward and fail metrics) on a set of held-out goal locations. We also compute the *half-trip time* ($T_{\frac{1}{2}}$), to reach halfway to the goal.

performance of the agents as the held-out area size increases. We believe that the performance drops on the large held-out areas (medium and coarse grid size) because the model cannot process new or unseen local landmark-based goal specifications, which is due to our landmark-based goal representation: as Figure 5 shows, some coarse grid held-out areas cover multiple landmarks. To gain further understanding, in addition to the *Test Reward* metric, we also use missed goals (*Fail*) and half-trip time ($T_{\frac{1}{2}}$) metrics. The missed goals metric measures the percentage of times goals were not reached. The half-trip time measures the number of agent steps necessary to cover half the distance separating the agent from the goal. While the agent misses more goal destinations on larger held-out grids, it still manages to travel half the distance to the goal within a similar time, which suggests that the agent has an approximate held-out goal representation that enables it to head towards it until it gets close to the goal and the representation is no longer useful for the final approach.

## 5.4 Transfer in Multi-city Experiments

A critical test for our proposed method is to demonstrate that it can provide a mechanism for transfer to new cities. By definition, the courier task requires a degree of memorization of the map, and what we focused on was not zero-shot transfer, but rather the capability of models to generalize quickly, learning to separate general ability from local knowledge when migrating to a new map. Our motivation for transfer learning experiments comes from the goal of continual learning, which is about learning new skills without forgetting older skills. As with humans, when our agent visits a new city we would expect it to have to learn a new set of landmarks, but not have to re-learn its visual representation, its behaviours, etc. Specifically, we expect the agent to take advantage of existing visual features (convnet) and movement primitives (policy LSTM). Therefore, using the *MultiCityNav* agent, we train on a number of cities (actually regions in New York City), freeze both the policy LSTM and the convolutional encoder, and then train a new locale-specific pathway (the goal LSTM) on a new city. The gradient that is computed by optimising the RL loss is passed through the policy LSTM without affecting it and then applied only to the new pathway.

We compare the performance using three different training regimes, illustrated in Fig. 6a: Training on only the target city (*single training*); training on multiple cities, including the target city, together (*joint training*); and joint training on all but the target city, followed by training on the target city with the rest of the architecture frozen (*pre-train and transfer*). In these experiments, we use the whole Manhattan environment as shown in Figure 1b, and consisting of the following regions "Wall Street", "NYU", "Midtown", "Central Park" and "Harlem". The target city is always the Wall Street environment, and we evaluate the effects of pre-training on 2, 3 or 4 of the other environments. We also compare performance if the skip connection between the convolutional encoder and the policy LSTM is removed.

We can see from the results in Figure 6b that not only is transfer possible, but that its effectiveness increases with the number of the regions the network is trained on. Remarkably, the agent that is pre-trained on 4 regions and then transferred to Wall Street achieves comparable performance to an agent trained jointly on all the regions, and only slightly worse than single-city training on Wall Street alone[3]. This result supports our intuition that training on a larger set of environments results in successful transfer. We also note that in the single-city scenario it is better to train an agent with a

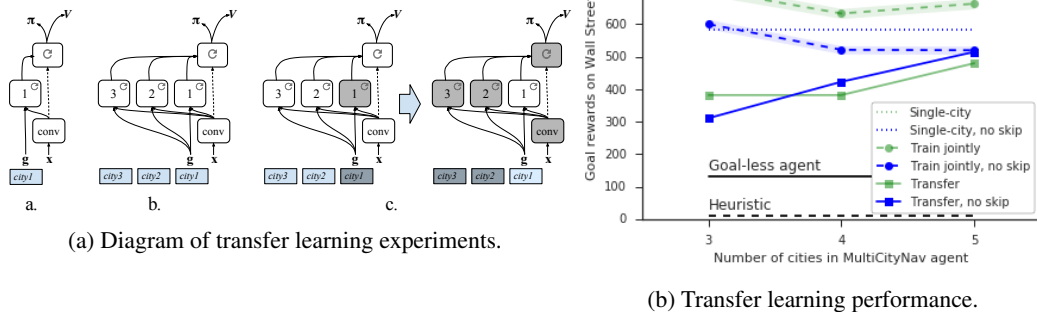

(a) Diagram of transfer learning experiments.

(b) Transfer learning performance.

Figure 6: Left: Illustration of training regimes: (a) training on a single city (equivalent to CityNav); (b) joint training over multiple cities with a dedicated per-city pathway and shared convolutional net and policy LSTM; (c) joint pre-training on a number of cities followed by training on a target city with convolutional net and policy LSTM frozen (only the target city pathway is optimised). Right: Joint multi-city training and transfer learning performance of variants of the *MultiCityNav* agent, evaluated only on the target city (Wall Street).

skip-connection, but this trend is reversed in the multi-city transfer scenario. We hypothesise that isolating the locale-specific LSTM as a bottleneck is more challenging but reduces overfitting of the convolutional features and enforces a more general interface to the policy LSTM. While the transfer learning performance of the agent is lower than the stronger agent trained jointly on all the areas, the agent significantly outperforms the baselines and demonstrates goal-dependent navigation.

## 6 Conclusion

Navigation is an important cognitive task that enables humans and animals to traverse a complex world without maps. We have presented a city-scale real-world environment for training RL navigation agents, introduced and analysed a new courier task, demonstrated that deep RL algorithms can be applied to problems involving large-scale real-world data, and presented a multi-city neural network agent architecture that demonstrates transfer to new environments. A multi-city version of the Street View based RL environment, with carefully processed images provided by Google Street View (i.e., blurred faces and license plates, with a mechanism for enforcing image take-down requests) has been released for Manhattan and Pittsburgh and is accessible from `http://streetlearn.cc` and `https://github.com/deepmind/streetlearn`. The project webpage at `http://streetlearn.cc` also contains resources on how to build and train an agent. Future work will involve learning landmarks from images and scaling up the navigation and path-planning thanks to hierarchical RL approaches.

## Acknowledgements

The authors wish to acknowledge Andras Banki-Horvath for open-sourcing the StreetLearn environment, Lasse Espeholt and Hubert Soyer for technical help with the IMPALA algorithm, Razvan Pascanu, Ross Goroshin, Pushmeet Kohli and Nando de Freitas for their feedback, Chloe Hillier, Razia Ahamed and Vishal Maini for help with the project, and the Google Street View team (Tilman Reinhardt, Wenfeng Li, Ben Mears, Karen Guo, Oliver Metzger, Jayanth Nayak) as well as Richard Ives and Ashwin Kakarla for their support in accessing the data.

## Footnotes

[1]`http://streetlearn.cc` (dataset) and `https://github.com/deepmind/streetlearn` (code).

[2]https://developers.google.com/maps/documentation/streetview/

[3]We observed that we could train a model jointly on 4 cities in fewer steps than when training 4 single-city models.

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
