[Supplementary Material · Supplement_LearningNavigateCitiesWithoutMap_NeurIPS2018_CameraReady.pdf]

# Learning to Navigate in Cities Without a Map
# (Supplementary Material)

**Piotr Mirowski, Matthew Koichi Grimes, Mateusz Malinowski, Karl Moritz Hermann,**
**Keith Anderson, Denis Teplyashin, Karen Simonyan, Koray Kavukcuoglu,**
**Andrew Zisserman, Raia Hadsell**
DeepMind
London, United Kingdom
{piotrmirowski, mkg, mateuszm, kmh, keithanderson, }@google.com
{teplyashin, simonyan, korayk, zisserman, raia}@google.com

## 1  Video of the Agent Trajectories and Observations

The video available at `http://streetlearn.cc` and `https://youtu.be/2yjWDNXYh5s` shows the performance of trained *CityNav* agents in the Paris Rive Gauche and Central London environments, as well as of the *MultiCityNav* agents trained jointly on 4 environments (Greenwich Village, Midtown, Central Park and Harlem) and then transferred to a fifth environment (Lower Manhattan). The video shows the high-resolution StreetView images (actual inputs to the network are $84 \times 84$ RGB observations), overlaid with the map of the environment indicating its position and the location of the goal.

## 2  Further Analysis

### 2.1  Architecture Ablation Analysis

Figure 1: Learning curves of the *CityNav* agent (2LSTM+Skip+HD) on NYU, comparing different ablations, all they way down to *GoalNav* (LSTM). 2LSTM architectures have a global pathway LSTM and a policy LSTM with optional Skip connection between the convnet and the policy LSTM. HD is the heading prediction auxiliary task.

In this analysis, we focus on agents trained with a two-day learning curriculum and early rewards at 200m, in the NYU environment. Here, we study how the learning benefits from various auxiliary tasks as well as we provide additional ablation studies where we investigate various design choices. We quantify the performance in terms of average reward per episode obtained at goal acquisition.

Each experiment was repeated 5 times with different seeds. We report average final reward and plot the mean and standard deviation of the reward statistic. As Fig. 1 shows, the auxiliary task of heading (*HD*) prediction helps in achieving better performance in the navigation task for the *GoalNav* architecture, and, in conjunction with a skip connection from the convnet to the policy LSTM, for the 2-LSTM architecture. The *CityNav* agent significantly outperforms our main baseline *GoalNav* (LSTM in Fig. 1), which is a goal-conditioned variant of the standard RL baseline [4]. *CityNav* perfoms on par with *GoalNav* with heading prediction, but the latter cannot adapt to new cities without re-training or adding city-specific components, whereas the *MultiCityNav* agent with locale-specific LSTM pathways can, as demonstrated in the paper's section on transfer learning. Our weakest baseline (*CityNav* no vision) performs poorly as the agent cannot exploit visual cues while performing navigation tasks. In our investigation, we do not consider other auxiliary tasks introduced in prior works [2, 3] as they are either unsuitable for our task, do not perform well, or require extra information that we consider too strong. Specifically, we did not implement the reward prediction auxiliary task on the convnet from [2], as the goal is not visible from pixels, and the motion model of the agent with widely changing visual input is not appropriate for the pixel control tasks in that same paper. From [3], we kept the 2-LSTM architecture and substituted depth prediction (which we cannot perform on this dataset) by heading and neighbor traversability prediction. We did not implement loop-closure prediction as it performed poorly in the original paper and uses privileged map information.

## 2.2 Goal Representation

Figure 2: Learning curves for *CityNav* agents with different goal representations: landmark-based, as well as latitude and longitude classification-based and regression-based.

As described in Section 3.1 of the paper, our task uses a goal description which is a vector of normalised distances to a set of fixed landmarks. Reducing the density of the landmarks to half, a quarter or an eighth (50%, 25%, 12.5%) does not significantly reduce the performance (Fig. 2). We also investigate some alternative representations: a) latitude and longitude scalar coordinates normalised to be between 0 and 1 (*Lat/long scalar* in Figure 2), and b) a binned representation *Lat/long binned* using 35 bins for X and Y each, with each bin covering 100m. The Lat/long scalar goal representations performs best.

Fig. 2 compares the performance of the *CityNav* agent for different goal representations $g_t$ on the NYU environment. The most intuitive one consists in normalized latitude and longitude coordinates, or in binned representation of latitude and longitude (we used 35 bins for X and 35 bins for Y, where each bin covers 100m, or 80 bins for each coordinate).

An alternative goal representation is expressed in terms of the distances $\{d^g_{t,k}\}_k$ from the goal position $(x^g_t, y^g_t)$ to a set of arbitrary landmarks $\{x_k, y_k\}_k$. We defined $g_{t,i} = \frac{\exp(-\alpha d^g_{t,k})}{\sum_k \exp(-\alpha d^g_{t,k})}$ and tuned $\alpha = 0.002$ using grid search. We manually defined 644 landmarks covering New York, Paris and London, which we use throughout the experiments and which are illustrated on Fig.2a. We observe that reducing the density of the landmarks to half, a quarter or an eighth has a slightly detrimental effect on performance because some areas are sparsely covered by landmarks. Because the landmark representation is independent of the coordinate system, we choose it and use it in all the other experimnets in this paper.

Finally, we also train a *Goal-less CityNav* agent by removing inputs $g_t$. The poor performance of this agent (Fig. 2) confirms that the performance of our method cannot be attributed to clever street graph exploration alone. *Goal-less CityNav* learns to run in circles of increasing radii—a reasonable, greedy behaviour that is a good baseline to the other agents.

Since the landmark-based representation performs well while being independent of the coordinate system and thus more scalable, we use this representation as canonical.

## 2.3 Allocentric and Egocentric Goal Representation

We do an analysis of the activations of the 256 hidden units of the region-specific LSTM, by training decoders (2-layer multi-layer perceptrons, MLP, with 128 units on the hidden layer and rectified nonlinearity transfer functions) for the allocentric position of the agent and of the goal as well as for the egocentric direction towards the goal. Allocentric decoders are multinomial classifiers over the joint Lat/Long position, and have $50 \times 50$ bins (London) or $35 \times 35$ bins (NYU), each bin covering an area of size 100m $\times$ 100m. The egocentric decoder has 16 orientation bins. Fig.3 illustrates the noisy decoding of the agent's position along 3 trajectories and the decoding of the goal (here, St Paul's), overlaid with the ground truth trajectories and goal location. The average error of the egocentric goal direction decoding is about $60°$ (as compared to $90°$ for a random predictor), suggesting some decoding but not a cartesian manifold representation in the hidden units of the LSTM.

Figure 3: Decoding of the agent position (blue dots) and goal position (cyan stars) over 3 trajectories (in red) with a goal at St Paul's Cathedral, in London (in black).

## 2.4 Reward Shaping: Goal Rewards vs. Rewards

The agent is considered to have reached the goal if it is within 100m of the goal, and the reward shaping consists in giving the agent early rewards as it is within 200m of the goal. Early rewards are shaped as following:

$$ r_t = \max\left(0, \min\left(1, \frac{200 - d_t^g}{100}\right)\right) \times r^g $$

where $d_t^g$ is the distance from the current position of the agent to the goal and $r^g$ is the reward that the agent will receive if it reaches the goal. Early rewards are given only once per panorama / node, and only if the distance $d_t^g$ to the goal is decreasing (in order to avoid the agent developing a behavior of harvesting early rewards around the goal rather than going directly towards the goal). However, depending on the path taken by the agent towards the goal, it could earn slightly more rewards if it takes a longer path to the goal rather than a shorter path. Throughout the paper, and for ease of comparison with experiments that include reward shaping, we report only the rewards at the goal destination (*goal rewards*).

# 3 Implementation Details

## 3.1 Neural Network Architecture

For all the experiments in the paper we use the standard vision model for Deep RL [4] with 2 convolutional layers followed by a fully connected layer. The baseline *GoalNav* architecture has a single recurrent layer (LSTM), from which we predict the policy and value function, similarly to [4].

The convolutional layers are as follows. The first convolutional layer has a kernel of size 8x8 and a stride of 4x4, and 16 feature maps. The second layer has a kernel of size 4x4 and a stride of 2x2, and 32 feature maps. The fully connected layer has 256 units, and outputs 256-dimensional visual features $f_t$. Rectified nonlinearities (ReLU) separate the layers.

The convnet is connected to the policy LSTM (in case of two-LSTM architectures, we call it a *Skip* connection). The policy LSTM has additional inputs: past reward $r_{t-1}$ and previous action $\mathbf{a}_{t-1}$ expressed as a one-hot vector of dimension 5 (one for each action: forward, turn left or right by $22.5°$, turn left or right by $67.5°$).

The goal information $g_t$ is provided as an extra input, either to the policy LSTM (*GoalNav* agent) or to each *goal* LSTM in the *CityNav* and *MultiCityNav* agents. In case of landmark-based goals, $g_t$ is a vector of 644 elements (see Section 4 for the complete list of landmark locations in the New York and London environments). In the case of Lat/Long scalars, $g_t$ is a 2-dimensional vector of Lat and Long coordinates normalized to be between 0 and 1 in the environment of interest. In the case of binned Lat/Long coordinates, we bin the normalized scalar coordinates using 35 bins for Lat and 35 bins for Long in the NYU environment, each bin representing 100m, and the vector $g_t$ contains 70 elements.

The goal LSTM also takes 256-dimensional inputs from to the convnet. The goal LSTM contains 256 hidden units and is followed by a *tanh* nonlinearity, a dropout layer with probability $p = 0.5$, then a 64-dimensional linear layer and finally a *tanh* layer. It is this (*CityNav*) or these (*MultiCityNav*) 64-dimensional outputs that are connected to the policy LSTM. We chose to use this bottleneck, consisting of a dropout, linear layer from 256 to 64, followed by a nonlinearity, in order to force the representations in the goal LSTM to be more robust to noise and to send only a small amount of information (possibly related to the egocentric position of the agent w.r.t. the goal) to the policy LSTM. Please note that the *CityNav* agent can still be trained to solve the navigation task without that layer.

Similarly to [4], the policy LSTM contains 256 hidden units, followed by two parallel layers: one linear layer going from 256 to 1 and outputing the value function, and one linear layer going from 256 to 5 (the number of actions), and a softmax nonlinearity, outputting the policy.

The heading $\theta_t$ prediction auxiliary task is done using an MPL with a hidden layer of 128 units, connected to the hidden units of each goal LSTM in the *CityNav* and *MultiCityNav* agents, and outputs a softmax of 16-dimensional vectors, corresponding to 16 binned directions towards North. The auxiliary task is optimized using a multinomial loss.

## 3.2 Learning Hyperparameters

The costs for all auxiliary heading prediction tasks, of the value prediction, of the entropy regularization and of the policy loss are added before being sent to the RMSProp gradient learning algorithm [5] (momentum 0, discounting factor 0.99, $\epsilon = 0.1$, initial learning rate 0.001). The weight of heading prediction is 1, the entropy cost is 0.004 and the value baseline weight is 0.5.

In all our experiments, we train our agent with IMPALA [1], an actor-critic implementation of deep reinforcement learning that decouples acting and learning. In our experiments, IMPALA results in similar performance to A3C [4] on a single city task, but as it has been demonstrated to handle better multi-task learning than A3C, we prefer it to A3C for our multi-city and transfer learning experiments. We use 256 actors for *CityNav* and 512 actors for *MultiCityNav*, with batch sizes of 256 or 512 respectively, and sequences are unrolled to length 50. We used a learning rate of 0.001, linearly annealed to 0 after 2B steps (NYU), 4B steps (London) or 8B steps (multi-city and transfer learning experiments). The discounting coefficient in the Bellman equation is 0.99. Rewards are clipped at 1 for the purpose of gradient calculations.

### 3.3 Curriculum Learning

Because of the distributed nature of the learning algorithm, it was easier to implement the duration of phase 1 and phase 2 of curriculum learning using the Wall clock of the actors and learners rather than by sharing the total number of steps with the actors, which explains why phase durations are expressed in terms of days, rather than in a given number of steps. With our software implementation, hardware and batch size as well as number of actors, the distributed learning algorithm runs at about 6000 environment steps/sec, and a day of training corresponds to about 500M steps. In terms of gradient steps, given than we use unrolls of length 50 steps and batch sizes of 256 or 512, each gradient step corresponds to either $50 \times 256 = 12800$ or $50 \times 512 = 25600$ environment steps, and is taken every 2s or 4s respectively for a speed of 6000 environment steps/sec.

## 4 Environment

For the experiments on data from Manhattan, New York, we relied on sub-areas of a larger StreetView graph that contains 256961 nodes and 266040 edges. We defined 5 areas by selecting a starting point at a given coordinate and collecting panoramas in a panorama adjacency graph using breadth-first-search, until a given depth of the search tree. We defined areas as following:

- Wall Street / Lower Manhattan: 6917 nodes and 7191 edges, 200-deep search tree starting at (40.705510, -74.013589).
- NYU / Greenwich Village: 17227 nodes and 17987 edges, 200-deep search tree starting at (40.731342, -73.996903).
- Midtown: 16185 nodes and 16723 edges, 200-deep search tree starting at (40.756889, -73.986147).
- Central Park: 10557 nodes and 10896 edges, 200-deep search tree starting at (40.773863, -73.971984).
- Harlem: 14589 nodes and 15099 edges, 220-deep search tree starting at (40.806379, -73.950124).

The Central London StreetView environment contains 24428 nodes and 25352 edges, and is defined by a bounding box between the following Lat/Long coordinates: (51.500567, -0.139157) and (51.526175, -0.080043). The Paris Rive Gauche environment contains 34026 nodes and 35475 edges, and is defined by a bounding box between Lat/Long coordinates: (48.839413, 2.2829247) and (48.866578, 2.3653221).

We provide, in a text file[1], the locations of the 644 landmarks used throughout the study.

## Footnotes

[1]Available at `http://streetlearn.cc`

[5] Tijmen Tieleman and Geoffrey Hinton. Lecture 6.5-rmsprop: Divide the gradient by a running average of its recent magnitude. *COURSERA: Neural networks for machine learning*, 4(2):26–31, 2012.