[Reviews · NeurIPS 2018]

Reviewer 1



This paper presents an impressive application of deep RL to navigation from large amounts of visual data, allowing an agent to navigate to a goal location from vision alone and no map. They further show that their proposed architectures which are motivated by the navigation task help learning, especially on multi-city tasks. The novelty of the paper lies mostly in the choice of architecture and particularly the relative success of the “transfer” setting using this architecture. This could deserve further inspection and experiments. In section 4.1 I am not sure if understand correctly what was meant by “general” vs. “locale-specific” and how they relate to the network modules. I believe “general” might mean things like how to navigate roads and buildings, while “locale-specific” means memorizing place locations. However, since x and g are seen by both LSTMs, how exactly they are operating seems unknowable. It would be good to make these points clearer and more concrete. In the experiments, freezing the weights for the “transfer” setting works alright with enough cities but performs worse than the single-city method, so this suggests the two types of information are not cleanly divided between the modules. While the architecture choices are justified by the performance gains, analysis of the learned features of each module would help motivate the architecture if possible. I could not find details of the exact reward function used. This is important information to include prominently in the paper. It is good that the reward shaping and curriculum results are included in the paper but they are not too surprising. Although the method is not very novel, there are some interesting architecture innovations and the results are very polished. Assuming the dataset and environment are released, it will be a very valuable testbed for the learning community and will likely prompt very interesting future research questions. The experiments are thorough and the writing is clear. Figure 2b image c: what is meant by the dotted line? Initially I thought this was the skip connection referred to in the text but it seems that skip connection also applies to CityNav. Figure 3a: why does GoalNav fail? It seems two things are changed from GoalNav to CityNav: the architecture and heading auxiliary prediction, so its tough to know which made the difference. Line 202: typo, missing word between “we in”. Line 218: sentence seems unfinished Great video!

Reviewer 2



This paper presents an approach for learning to navigate in cities without a map. Their approach leverages existing deep reinforcement learning approaches to learn to navigate to goals using first-person view images, in which the goals are defined using local landmarks. The authors propose a neural network architecture, in which the goal is processed separately. Extensive experiments are conducted on real-world Google streetview datasets, showing the approach works and which aspects of the approach are important. First, I want to be explicit: (1) I reviewed this paper for ICML 2018 (2) I now know the authors due to a widely publicized blog post Overall, this paper is well written, clear, and has extensive experiments. Compared to the ICML submission, the contributions are well-grounded with respect to related work, and accurately show that this paper is the (non-trivial) combination of prior works. My main critique is the lack of novelty. However, given the strong motivation, clear writing, and extensive experiments, I think this paper could spearhead a good direction for learning for navigation. However, re-implementing this paper would be a huge burden that would stop many from even trying. Although the authors state they will release the code before the conference, it has been over 3 months since the ICML submission and no code has been released. Therefore I’ll say the following: if the authors anonymously publish their code to a private github repo with instructions on how to download and run one of their experiments, I will change my overall score to an 8 and will argue strongly for acceptance. Minor comments: (1) typo line 202 (2) Regarding section 4.2, it is correct that while the courier task is similar to Montezuma’s in that both are sparse reward, it is easy to define a curriculum for the courier task and not Montezuma’s. However, the discussion lacks the following subtlety: if you can define a curriculum, you can also define a shaped reward; a trivial example would be rewards for entering a new curriculum. Therefore the motivation for having sparse rewards for the courier task is slightly artificial. This discussion is probably too nuanced for the paper.

Reviewer 3



This paper presents an application of a deep reinforcement learning model to a new navigation task based on real-world images taken from Google Street View. The paper is well written and easy to read. The task is interesting, the results are impressive and show that an external memory is not even required in order to achieve a good performance. My main concerns are that: a) The model cannot work on maps it has not seen during training, and requires to be fine-tuned on new maps. Also, the point of doing transfer between multiple cities does not seem obvious, as the model still needs to be trained (tuned) on the new map and does not perform as well as a single-city model. b) The model is evaluated on the training maps themselves, but using held-out areas. The size of these held-out areas seem to have a significant impact on the model which suggests that the model is overfitting on the training maps. A couple of comments / questions: 1) I’m not sure I understood how the landmarks are selected to define the goal, even after looking at the Appendix. Are the landmarks fixed and always the same across a city? The example Figure 2. a) shows an example with 5 landmarks used to represent the location of a particular goal, and I initially thought that given a location, g_t was computed using some nearby landmarks. But it turns out that the landmarks you consider are always the same (all of them) and not only the ones nearby? 2) I found line 138 "the agent receives a higher reward for longer journeys" a bit confusing. It sounds like the agent is given more reward for taking paths longer than necessary. It is not totally clear to me why you would want to give more reward for distant goals (in the end you want to find distant goals as much as the nearby ones). Instead I would have found it natural to give a higher reward to the agent if it is able to find the goal quickly. 3) Section 4.2 suggests that curriculum learning is an important part of the training procedure, but Figure 3) c) indicates that ER alone (with the appropriate range) converges faster and to a better final performance. Is that right? Any reason to keep using curriculum learning for this task? 4) I was surprised by how much the auxiliary task of heading is helping the model given the simplicity of this task. Any intuition of why it helps that much? Doesn’t the loss of this task converge to zero very quickly? 5) Figure 5: did you observe that during training, the agent tends to avoid the held-out areas naturally? Or, given a random goal location, it will navigate through it using the held-out areas as much as the other ones? The gap of performance between “fine” and “coarse” grid size is quite significant and I’m wondering if this can be an issue resulting from the agent not being familiar with the held-out areas at all. 6) When the model is trained on some cities and fine-tuned on a new city, how long does it take to fine tune it compared to training a model from scratch on the new city alone? I think this is an important result to know. The experiments show that a single-city network performs better than the ones trained jointly and fine-tuned, so if fine-tuning is not faster, I don’t see the point in training anything else than single-city networks. 7) Line 74 and line 172, Lample & Chaplot should be cited along with Mirowski et al. and Jaderberg et al., as they were the firsts to publish results showing the benefits of training with auxiliary tasks.